# Assessment of Air Pollution in Different Areas (Urban, Suburban, and Rural) in Slovenia from 2017 to 2021

**Maja Ivanovski [1,2,*], Kris Alatič [2], Danijela Urbancl [1], Marjana Simonič [1], Darko Goričanec [1] and Rudi Vončina [2]**

[1] Faculty of Chemistry and Chemical Engineering, University of Maribor, Smetanova 17, 2000 Maribor, Slovenia
[2] Milan Vidmar Electric Power Research Institute, Department for Environment, Hajdrihova 2, 1000 Ljubljana, Slovenia
[*] Correspondence: maja.ivanovski@eimv.si

**Abstract:** Air pollution can have a significant effect on human health. The present work is focused on the investigation of daily, monthly, and annual concentration levels of five typical air pollutants ($SO_2$, $NO_2$, $NO_X$, $PM_{10}$, and $PM_{2.5}$) in the Republic of Slovenia (RS) from January 2017 to December 2021. The study was conducted at five different monitoring stations of the following kind: traffic (A), industrial (D), and background (B, C, E). The obtained results showed a decline in the average concentrations for all the studied air pollutants through the years, respectively. The daily average $SO_2$ concentrations were the lowest in the year 2021 at location B, which is classified as background location, while the highest were detected in the year 2018 at location E, which is also classified as background location. The average daily concentrations of $NO_2$ and $NO_X$ were the highest at location A in the year 2017, whereas the lowest were detected in the year 2010 and 2021. It is believed that those results are a consequence of measures set by the Slovenian government during the COVID-19 pandemic. The $PM_{10}$ and $PM_{2.5}$ daily average concentrations were the highest at location A in 2017, while the lowest were observed in the year 2019 at location C. Meteorological parameters (temperature, wind speed, and relative humidity) were studied in addition. In general, the high temperatures in ambient air are responsible for the intense concentrations of air pollutants. It was found in the study results for temperature, wind speed, and relative humidity that no significant difference was shown between studied years.

**Keywords:** air pollution; air quality; air pollutants; meteorology; Republic of Slovenia



## 1. Introduction

Air pollution is considered as one of the most important public health problems related to environmental pollution. It represents a major global threat that is practically impossible to avoid, especially in developing countries. According to the European Environmental Agency (EEA) and the World Health Organization (WHO, Geneva, Switzerland), in the last decade, air pollution has become the second largest environmental issue, just after climate change [1]. Every year, approximately 7 million people die because of it, from which roughly 4.9 million deaths are caused by urban ambient air [1].

Lately, there have been critically high concentrations of air pollutants measured in many large European (EU) cities, affecting the quality of life and public health [2]. Among all air pollutants, sulfur dioxide ($SO_2$), nitrogen oxides ($NO_2$/$NO_X$), ozone ($O_3$), carbon oxides (CO, $CO_2$), particulate matter (PM), and volatile organic compounds (VOCs) are considered as some of the most common anthropogenic air pollutants caused by heavy road traffic, domestic heating, and local industry [3,4]. Years ago, $SO_2$ was declared as the largest environmental problem in the EU caused by the production and co-production of electricity and thermal energy, industrial processes, and the usage of coal for domestic heating in households, but, today, $NO_2$/$NO_X$, $CO_2$, and, especially, particulate matter (PM) are taking the lead [5]. It has been stated by the EEA that, between the years 2005

and 2019, emissions of the abovementioned pollutants declined significantly in most EU countries. For example, $SO_X$ emissions decreased by 76%, $NO_X$ by 42%, VOCs by 29%, and $PM_{2.5}$ by 29% [6]. The cause for such a decline in emissions is partly due to a decrease in emissions from the energy, industry and transport sectors, and partly due to the laws and standards adopted, legalized, and updated through the years in the EU regarding air pollution, such as the "Large Combustion Plant Directive (EC/80/2001)", the "Industrial Emissions Directive (EC/75/2010)", "European Pollutant Release and Transfer Register Regulation (EC/166/2006)", "Directive on Ambient Air Quality and Cleaner Air for Europe (EC/50/2008)", and the "Directive on heavy metals and polycyclic aromatic hydrocarbons in ambient air EC/107/2004)".

In addition, not long ago, almost all countries revised their air quality guidelines world-wide, and proposed new regulations according to the guidelines published by the WHO in 2021 [7]. The guidelines were revised as a result of the growing threat of air pollution on population health [8–11], and contain information regarding limit values of air pollutants, which are generally lower compared to those in the 2005 version [12]. The following differences could be observed: for $SO_2$, the daily limit value increased from 20 $\mu g/m^3$ in 2005 to 40 $\mu g/m^3$ in the 2021 guidelines, while the daily limit value for the $NO_2$ parameter in the 2005 guidelines was not determined, whereas the new regulations proposed a daily limit value of 25 $\mu g/m^3$. An annual limit value for the $NO_2$ parameter decreased from 40 $\mu g/m^3$ set in the 2005 guidelines to 10 $\mu g/m^3$ in the 2021 guidelines. For $PM_{10}$, the daily limit value of 50 $\mu g/m^3$ decreased to 45 $\mu g/m^3$, and, in the same way, the annual limit value decreased from 20 $\mu g/m^3$ to 15 $\mu g/m^3$. A similar value was obtained for the $PM_{2.5}$ parameter. The daily limit value of 25 $\mu g/m^3$ set in 2005 decreased to 15 $\mu g/m^3$ in 2021, and, consequently, the annual limit value decreased from 10 $\mu g/m^3$ to 5 $\mu g/m^3$.

Systematic measurements of the concentrations of air pollutants at permanent measuring points in the Republic of Slovenia (RS) began in the mid-1970s by the Slovenian Environment Agency (ARSO) and have since provided basic air quality data [13]. The measuring network in the RS currently consists of 27 ground-based nationwide monitoring stations, with 23 more monitoring stations owned by industrial companies or municipalities. These monitoring stations are based in eight different regions, thirteen different municipalities, and are divided by area (urban, suburban, and rural) and measurement (traffic, background, and industrial location) type [14]. It has been stated by ARSO and the National Institute for Public Health (NIJZ), that recently, the RS has currently been experiencing excessive air pollution [15].

Air pollution in the country is generally caused by traffic and individual fireplaces, as well as by industry (coal burning for energy generation, industrial emissions, waste burning, construction activities) [15]. It is the terrain that makes air pollution in the country more susceptible to weather variability, especially during winter time when there are temperature inversions [3]. ARSO additionally proposed national regulations regarding air quality, which mostly coincide with those proposed by the WHO. For $SO_2$, the daily limit value is set to 125 $\mu g/m^3$ and must not be exceeded for more than three times per year. For $NO_2$, the daily limit value is not determined, whereas the annual limit value is set to 40 $\mu g/m^3$. For $PM_{10}$, the daily limit value is 50 $\mu g/m^3$ and must not be exceeded 35 times per year, while an annual limit value is set to 40 $\mu g/m^3$. Lastly, the daily limit value for $PM_{2.5}$ parameter is 25 $\mu g/m^3$ (must not be exceeded 24 times per year) and the annual limit value decreased to 20 $\mu g/m^3$ [5]. The regulations are currently valid.

In such a context, the main goal of this work is to determine, analyze, and compare the concentrations of some basic air pollutants ($SO_2$, $NO_2/NO_X$, $PM_{10}$, and $PM_{2.5}$) in the ambient air at five sampling locations, characterized as traffic, industrial, and background, based in the Republic of Slovenia (RS), between the years 2017 and 2021 to determine the differences between them. These years were chosen because they represent milestones in analyzing air quality in the RS, as the years 2020 and 2021 were marked with the COVID-19 pandemic. Previous years, 2017–2019, were included in the study to obtain comparative results. The chosen sampling locations present a region diversity in the RS.

Related meteorological parameters (temperature, relative humidity, wind speed, and wind direction) were studied as well.

## 2. Methodology

### 2.1. Study Area

The RS is located in Central Europe, at 46.15° N and 46.08° E, with an average of 492 m above mean sea level. It is located in the southeastern part of the Alps, surrounded by the mountains, forests, and Adriatic sea. The climate is influenced by a variety of geographical relief—there are fewer lowlands, valleys, and basins, which are surrounded by hills. There are three typical climates in the country: a continental climate with a stark difference in winter and summer temperatures in the northeast; a sub-Mediterranean climate in the coastal region; and a severe Alpine climate present in the high mountain regions. Precipitation, often coming from the Gulf of Genoa, varies across the country as well, with over 3.500 mm in some western regions and dropping down to 800 mm in Prekmurje [14]. The winds (north, northeast, and southeast winds) are mainly weak, especially on the ground, due to the characteristic leeward position.

National and local meteorology and relief variability of the surface are closely related to the concentration of emissions in the air, so for a comprehensive insight into the state of ambient air quality in the environment, it is necessary to monitor meteorological parameters as well [16]. For instance, due to the typical temperature inversions in the cold parts of the calendar year, emissions of harmful substances can remain longer in the air and can therefore cause a lot of air pollution within a relatively small volume of air [17]. This situation is most problematic in populated valleys and basins, where the population density is the highest and discharges are most common. In addition, national and local meteorology depends on the relief diversity in the environment, as it mainly affects the movement of air masses. In the case of favorable meteorological conditions, emissions can travel long distances and thus affect a larger area [16].

Therefore, the study of $SO_2$, $NO_2/NO_X$, $PM_{10}$, and $PM_{2.5}$ pollution characteristics in the RS is of great significance for atmospheric pollution management in the EU.

### 2.2. Location Description

This study was conducted in five different monitoring stations (Figure 1). The first monitoring station, hereinafter defined as sampling location A, is located in the heart of the RS, in the capital city Ljubljana. The station represents a favorable sampling site due to its central position in the city center (1000 m distant) surrounded by buildings of different sizes. The monitoring station is located very close to a road with significant daily traffic (10 m distant). In addition, the capital city is the geographical, cultural, economical, political, and administrative center of the country, an important transport hub, with a junction of international motorways running in four directions (crossing two corridors-directions) and railway lines (five to six directions). The second monitoring station, hereinafter defined as sampling location B, is located in the north-east of the RS, at the confluence of the Savinja and Voglajna rivers in the Lower Savinja Valley, approximately 3000 m distant from the city center. This monitoring station is surrounded by industrial facilities, such as a thermal power plant (around 4000 m distant) or metallurgical-chemical facility (around 4000 m). The third monitoring station, hereinafter defined as sampling location C, is located in the northern part of the RS, approximately 13,000 m distant from the capital city. The monitoring station is located close to the primary school, local hospital, and sports facilities (around 500 m distant). The fourth monitoring station, hereinafter defined as sampling location D, is located in the north-east of the RS in the valley along the Paka river, with a thermal power plant distant only by 500 m. Lastly, the fifth monitoring station, hereinafter defined as sampling location E, is located in the south-east of the RS, around 6000 m away from the nuclear power plant. The monitoring station is located in the typical rural area of the country.

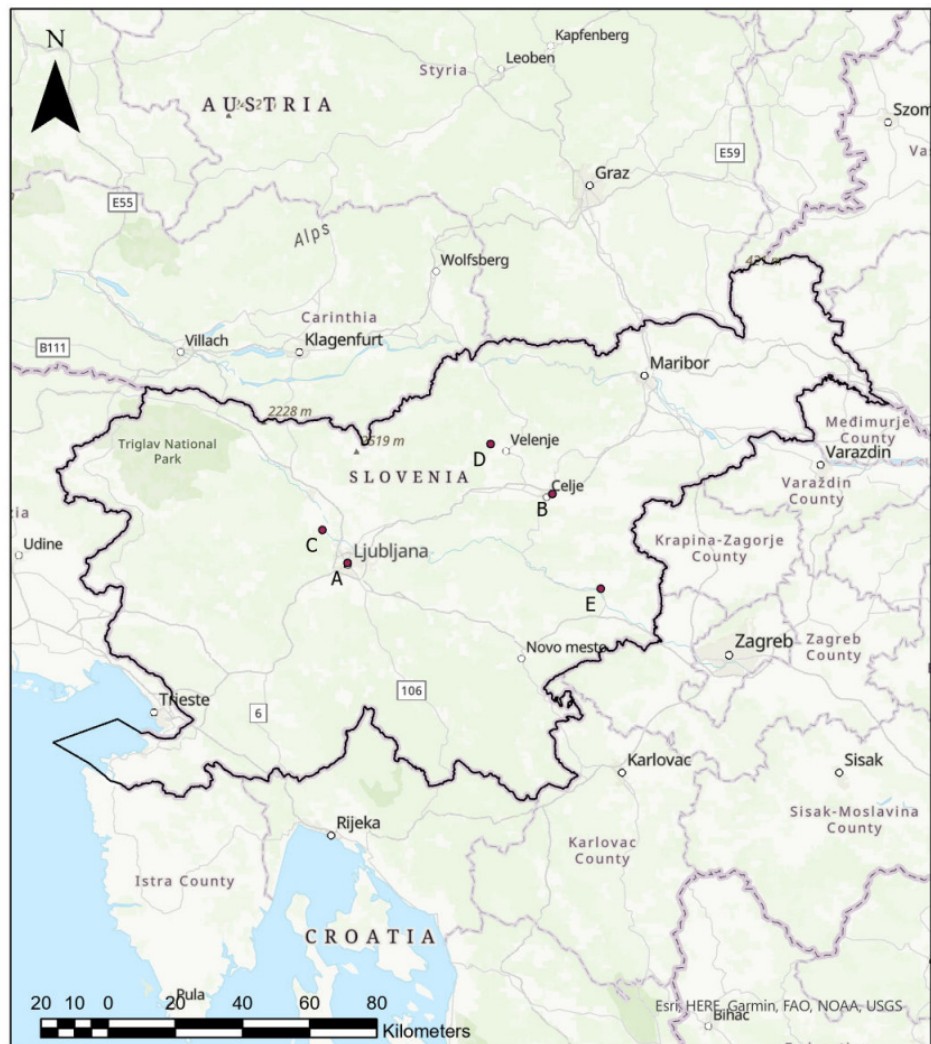

**Figure 1.** The geographical location of the study area with marked monitoring stations (A—traffic, B—background, C—background, D—industrial, and E—background).

All monitoring stations are located approximately 1 m above the ground, and are equipped with air condition and ventilation. Table 1 shows the characteristics of studied sampling locations. The classification and description of area (urban, suburban, and rural) and measurement (traffic, industrial, background) type is summarized as in the official national document stated by Bec et al. [17]. The RS has a typical spatial dispersion of settlement. Compared to other European cities, Slovenian cities are small- to medium-sized. Out of a total of 6.035 Slovenian settlements, 90% of settlements have less than 500 inhabitants and only two settlements have more than 50,000 inhabitants (Ljubljana and Maribor). There are three types of regions in the RS: urban, suburban, and rural. Urban area type indicates the area of a city or urban settlement with nearby urbanized surroundings, which differs from the more distant hinterland, mainly in terms of higher population density, compact construction, and a small share of the farming population. Suburban area type indicates an area type that is increasingly spreading to the countryside, with visible effects on the changed social-geographical, economic-geographical, and physical images of settlements. Meanwhile, rural area type indicates typical agricultural areas. The monitoring station with classification »traffic« is the one that measures air pollution near the ground level along the street, the »industrial« monitoring station is the one that measures close to the industrial areas, such as industrial pants, power plants, or thermal power plants, while the »background« monitoring station is the one that usually measures tens of meters in the air.

**Table 1.** Altitude and coordinates of the air quality monitoring stations.

| Sampling Location | Altitude (m) | $G_{KKY}$ | $G_{KKX}$ | Area Type | Measurement Type | Pollutants Measured | Meteorological Data Measured |
|---|---|---|---|---|---|---|---|
| A | 299 | 461,919.00 | 101,581.00 | Urban | Traffic | $SO_2$, $NO_2$, $NO_X$, $PM_{10}$, and $PM_{2.5}$ | |
| B | 240 | 522,888.00 | 122,129.00 | Urban | Background | $SO_2$, $NO_2$, $NO_X$ | Air temperature (°C), |
| C | 346 | 454,441.00 | 111,411.00 | Suburban | Background | $PM_{10}$ | Wind direction and speed (m/s), |
| D | 362 | 504,504.00 | 137,017.00 | Suburban | Industrial | $SO_2$, $NO_2$, $NO_X$, $PM_{10}$, and $PM_{2.5}$ | Humidity (%) |
| E | 390 | 537,299.00 | 93,935.00 | Rural | Background | $SO_2$, $NO_2$, $NO_X$ | |

### 2.3. Data Sources and Collection

The air pollutants data and meteorological data were obtained from the Milan Vidmar Electric Power Research Institute (EIMV), the Department for the Environment, and the Slovenian Environment Agency (ARSO). Real time data from the monitoring stations are available on open access to the public on the following platform: https://www.arso.gov.si/zrak/kakovost%20zraka/podatki/ (accessed on 15 March 2023).

The concentrations of the five basic air pollutants, $SO_2$, $NO_2$/$NO_X$, $PM_{10}$, and $PM_{2.5}$, were collected from January 2017 to December 2021 at the above-described measuring locations, respectively. $SO_2$ was measured using a UV fluorescence-based instrument (Teledyne API, model 100E), $NO_2$ and $NO_X$ were determined using a chemiluminiscence analyzer (Teledyne API, model 200E), and $PM_{10}$ and $PM_{2.5}$ were analyzed using an automatic gravimetric instrument (Sven Leckel, model SEQ47/50). Pre-weighed (Whatman™, diameter 47 mm, porosity 8.0 and 0.4 µm) and prebaked (850 °C, 3 h) fiber filters have been used. Before and after the sampling, the filters were placed in a room with controlled humidity $50 \pm 5\%$ and temperature $25 \pm 2$ °C, for at least 48 h before being weighed using a Mettler Toledo analytical balance, model AG 245, with a resolution of 0.01 mg. The mass concentration of particles in the air was expressed in µg/m³. The mass determination was performed under repeatability conditions ($n > 3$), aiming at minimizing the mass uncertainty. Outdoor air quality analyzers were installed in the measuring station, which was equipped with air conditioning and communication equipment.

In addition, real time meteorological data on the daily average temperature (T), relative humidity (RH), wind direction (WD), and wind speed (WS) during the study period at all measuring locations were obtained using the following monitoring equipment: the measurements of air temperature were performed with a resistance thermometer (Thermometer™, model RS232), relative humidity measurements were performed with a capacitive encoder, which, with the help of an electronic circuit line, raised and amplified the changes in humidity in the air, and converted them into a suitable analogue electrical output signal (Vaisala Weather Transmitter, model WXT520), and finally, the measurements of wind direction and wind speed were performed with an ultrasonic anemometer (Vaisala Weather Transmitter WXT520).

During the study periods, in order to determine conformity, all measuring devices were scientifically managed and maintained to ensure adequate control over the measured values and conditions, and to avid any possible contamination [18].

### 2.4. Evaluation Method

The hourly values were averaged for each day to obtain average daily concentrations for each studied pollutant at each location. Average monthly and yearly concentrations were studied as well. For each pollutant, the average data are discussed based on location and studied year (2017 and 2021, respectively). The minimum and maximum values are added with standard deviations. It must be taken into account that not all air pollutants ($SO_2$, $NO_2$/$NO_X$, $PM_{10}$, or $PM_{2.5}$) are measured at all studied locations during all study periods. For example, on location C, only parameter $PM_{10}$ was measured through the years; therefore, obtained results are discussed only for this parameter at this location. Furthermore, the daily average concentrations of all basic air pollutants were calculated

only when there were more than 16 h of valid data [19]. Furthermore, the associations between $NO_2$ and $NO_X$, and $PM_{10}$ and $PM_{2.5}$ were investigated using Pearson's correlation coefficient. The results are presented separately for nitrogen oxides and particulate matter, based on the location and studied year. Finally, the meteorological parameters were investigated.

Quality control was carried out daily to ensure reliability of the monitoring data. All measuring devices were automatically calibrated according to the international calibration standard EN ISO/IEC 17025:2017.

## 3. Results and Discussion

### 3.1. SO₂ Concentrations between 2017 and 2021

A few decades ago, $SO_2$ was the largest environmental issue near cities and thermal power plants in the RS, where the largest sources of emissions at that time were energy production, industry, and coal burning in individual furnaces. By slowly abandoning coal and by building desulfurization devices and wastewater treatment plants, the emissions of $SO_2$ decreased so much that its concentration levels have for several years now been below the lower limit value [17].

As already mentioned, according to the latest national air quality standards [17], the limit value for the daily maximum moving average of $SO_2$ in the RS is 125 $\mu g/m^3$ and must not be exceeded by more than three times per year; for the hourly value, the limit value is set to 350 $\mu g/m^3$ and must not be exceeded by more than 24 times per year. In the RS, the daily limit value is higher than the daily limit value stated in WHO guidelines.

This study is predominantly focused on the analyzing air quality during the years between 2017 and 2021 as these years represent a mark in analyzing air quality in the RS—in 2020, the COVID-19 pandemic occurred. In addition, previous years (2017, 2018, 2019) and after (2021) have been included to better understand the changes in air quality throughout the studied years. $SO_2$ parameter was studied at four locations (A, B, D, E). During all studied years and during the monitoring periods, the $SO_2$ data availability was more than 90% at all locations. The plots for daily average concentrations at all locations during the study periods are in the Supplementary Materials (SM) (Figure S1). Average, minimum, and maximum values with standard deviations for all studied years for each pollutant were additionally added (SM, Table S1). From the daily average data, it can be observed that the $SO_2$ concentrations were the lowest at location B in the year 2021 (1.28 $\pm$ 0.06 $\mu g/m^3$), while the highest were at location E in the year 2018 (21.56 $\pm$ 1.08 $\mu g/m^3$). Location E is located near the national nuclear power plant; therefore, the obtained average value was expected. In that year, a relatively high average daily value was also detected at location D, which is located close to the largest thermal power plant in the RS (19.45 $\pm$ 0.97 $\mu g/m^3$). Furthermore, the limit value of daily $SO_2$ concentration levels at location A was exceeded by seven times in 2017, three times in 2018, sixteen times in 2019, nine times in 2020, and seven times in 2021; at location D, the concentration was exceeded by 16 times in 2017, 19 times in 2018, 10 times in 2019, 14 times in 2020, and 9 times in 2021; and at location E, the concentration was exceeded 22 times in 2017, 14 times in 2018, 12 times in 2019, 14 times in 2020, and 12 times in 2021. At location B, the daily limit concentration value was not exceeded.

Figure 2 shows the average monthly $SO_2$ concentrations at studied locations throughout the years, while Figure 3 shows the average yearly $SO_2$ concentrations at studied locations during the same study period. Average monthly $SO_2$ concentrations at location A were 1.58 $\pm$ 0.08 $\mu g/m^3$ in 2017, 1.16 $\pm$ 0.06 $\mu g/m^3$ in 2018, 4.66 $\pm$ 0.23 $\mu g/m^3$ in 2019, 4.16 $\pm$ 0.21 $\mu g/m^3$ in 2020, and 2.17 $\pm$ 0.11 $\mu g/m^3$ in 2021. The highest average monthly $SO_2$ concentration was in the year 2019 (4.66 $\pm$ 0.23 $\mu g/m^3$), whereas the lowest was in the year 2018 (1.16 $\pm$ 0.06 $\mu g/m^3$). In January 2019, the highest average monthly concentration was detected (9.00 $\pm$ 0.45 $\mu g/m^3$), whereas the lowest average monthly concentration was detected in May and June (3.00 $\pm$ 0.15 $\mu g/m^3$). In 2018, the highest average monthly concentration of 2.00 $\pm$ 0.10 $\mu g/m^3$ was detected in January, September, October, and in November, whereas in other months, the average monthly $SO_2$ concen-

trations varied between $0.00 \pm 0.00$ µg/m$^3$ and $1.00 \pm 0.05$ µg/m$^3$. Average monthly SO$_2$ concentrations at location B were $5.17 \pm 0.26$ µg/m$^3$ in 2017, $6.08 \pm 0.30$ µg/m$^3$ in 2018, $8.17 \pm 0.41$ µg/m$^3$ in 2019, $10.25 \pm 0.51$ µg/m$^3$ in 2020, and $1.33 \pm 0.07$ µg/m$^3$ in 2021. The highest average monthly SO$_2$ concentration at location B was in the year 2020 ($10.25 \pm 0.51$ µg/m$^3$), whereas the lowest was in the year 2021 ($1.33 \pm 0.07$ µg/m$^3$). In 2020, the highest average monthly concentration was detected in April ($16.00 \pm 0.80$ µg/m$^3$), whereas the lowest average monthly concentration was detected in December ($2.00 \pm 0.10$ µg/m$^3$). In 2021, the highest average monthly concentration of $3.00 \pm 0.15$ µg/m$^3$ was detected in January, whereas during other months the average monthly concentration varied from $0.00 \pm 0.00$ µg/m$^3$ to $2.00 \pm 0.10$ µg/m$^3$. Average monthly SO$_2$ concentrations at location D were $19.92 \pm 0.99$ µg/m$^3$ in 2017, $11.92 \pm 0.60$ µg/m$^3$ in 2018, $9.75 \pm 0.49$ µg/m$^3$ in 2019, $10.42 \pm 0.52$ µg/m$^3$ in 2020, and $10.83 \pm 0.54$ µg/m$^3$ in 2021. The highest average monthly SO$_2$ concentration was in the year 2017 ($19.92 \pm 0.99$ µg/m$^3$), whereas the lowest was in the year 2019 ($9.75 \pm 0.49$ µg/m$^3$). In 2017, the highest average monthly concentration was detected in March and April ($26.00 \pm 1.30$ µg/m$^3$), and the lowest average monthly concentration was detected in September ($14.00 \pm 0.70$ µg/m$^3$). In 2019, the highest average monthly concentration was detected in January and February ($16.00 \pm 0.80$ µg/m$^3$) and the lowest average monthly concentration was detected in May ($5.00 \pm 0.25$ µg/m$^3$). Average monthly SO$_2$ concentrations at location E were $3.83 \pm 0.19$ µg/m$^3$ in 2017, $5.08 \pm 0.25$ µg/m$^3$ in 2018, $5.58 \pm 0.28$ µg/m$^3$ in 2019, $7.42 \pm 0.37$ µg/m$^3$ in 2020, and $4.41 \pm 0.22$ µg/m$^3$ in 2021. The highest average monthly SO$_2$ concentration was in the year 2020 ($7.42 \pm 0.37$ µg/m$^3$), whereas the lowest was in the year 2017 ($3.83 \pm 0.19$). In 2020, the highest average monthly concentration was detected in June and September ($10.00 \pm 0.50$ µg/m$^3$) and the lowest average monthly concentration was detected in January and February ($5.00 \pm 0.25$ µg/m$^3$). In 2017, the highest average monthly concentration was detected in January ($1.00 \pm 0.30$ µg/m$^3$), while the lowest average monthly concentration was detected in December ($2.00 \pm 0.10$ µg/m$^3$).

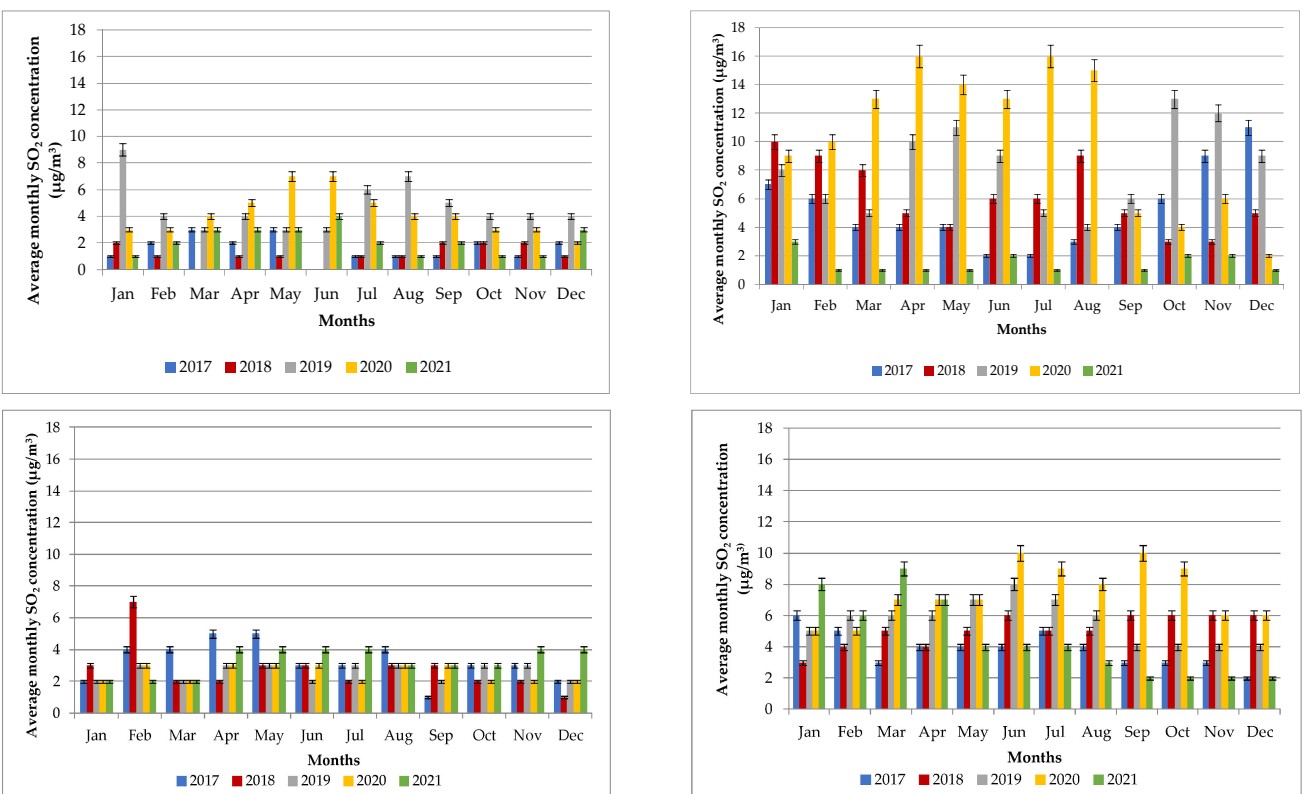

**Figure 2.** Average monthly SO$_2$ concentrations at locations A, B, D, and E between 2017 and 2021 (from left to right, from top to bottom).

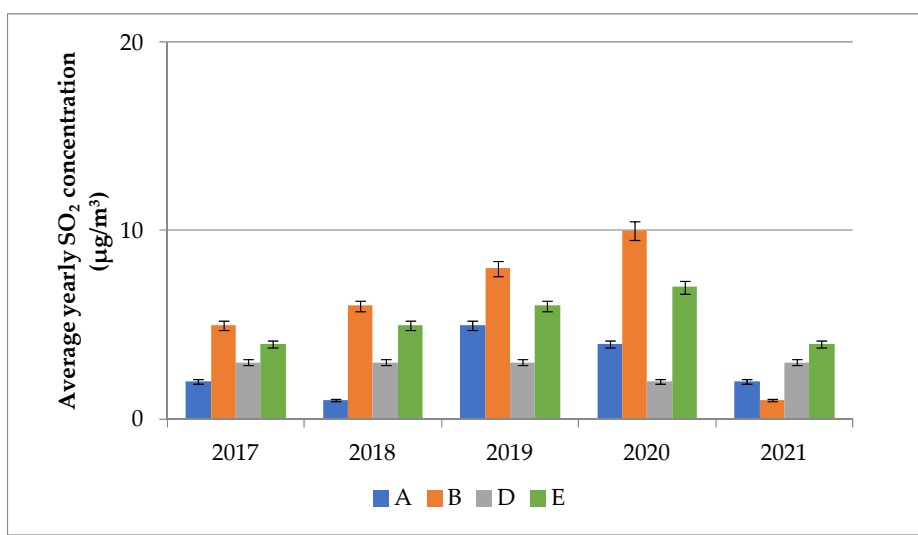

**Figure 3.** Average yearly SO$_2$ concentrations at locations A, B, D, and E between 2017 and 2021.

Average annual SO$_2$ concentrations were the highest at location B ($10.00 \pm 0.50$ μg/m$^3$) in 2020, while the lowest were detected at location A ($1.00 \pm 0.05$ μg/m$^3$) in 2018. For location B, the average annual mass concentrations for the SO$_2$ parameter exceeded the critical limit value for vegetation protection (20 μg/m$^3$) by five times in 2017, six times in 2018, eight times in 2019, ten times in 2020, and one time in 2021. For location D, the average annual mass concentration was exceeded by three times in 2017, 2018, 2019, and 2021, and two times in 2020. For location E, the average annual mass concentration was exceeded by four times in 2017, five times in 2018, six times in 2019, seven times in 2020, and four times in 2021. There were no exceeding annual mass concentrations at location A through the years.

In general, average daily, monthly, and annual SO$_2$ concentrations were nearly the same through all the studied years. SO$_2$ concentration values have not changed drastically over the studied years, even though some anomalies occurred, which could be connected to the local industrial, meteorological, or other unknown factors. As stated in the work of Calkins et al. [20], the distribution of atmospheric SO$_2$ is not only dependent on the emission of SO$_2$, but also by meteorological conditions. It was similarly stated in the work of Chen et al. [21]. At the measuring locations around the thermal power plant and nuclear power plant, some differences were recognized as a result of operation hours and meteorological conditions [13]. The obvious decrease in concentration occurred after the installation of desulfurization devices on individual blocks of thermal power plants [13].

*3.2. NO$_2$ and NO$_X$ Concentrations between 2017 and 2021*

More than half of NO$_X$ emissions in the atmosphere come from transportation traffic [22]. According to the ARSO, the limit value for the hourly maximum average of NO$_2$ in the RS is 200 μg/m$^3$ and must not be exceeded more than 18 times per year; annual average limit value is set to 40 μg/m$^3$ [17]. The same values are proposed by the WHO 2021 guidelines [7].

In this work, NO$_2$ parameter was studied from 2017 to 2021 at four locations (A, B, D, E). The data availability was more than 90% at all locations. The plots for daily average concentrations at all locations during the study period are in the SM (Figure S2). Average, minimum, and maximum values with standard deviations for all studied years for each pollutant are included (SM, Table S2). From the daily average data, it can be observed that the NO$_2$ concentrations were the lowest at location E in the year 2019 ($5.05 \pm 0.25$ μg/m$^3$), while the highest were at location A in the year 2017 ($550.23 \pm 2.51$ μg/m$^3$). Location A is located very close to a road with significant daily traffic, whereas location E represents a

monitoring station located near the national nuclear power plant; therefore, the obtained average values were expected.

Figure 4 shows average monthly $NO_2$ concentrations at studied locations throughout the years. Average monthly $NO_2$ concentrations at location A were $111.92 \pm 5.97$ µg/m³ in 2017, $34.00 \pm 1.70$ µg/m³ in 2018, $41.08 \pm 2.05$ µg/m³ in 2019, $35.16 \pm 1.76$ µg/m³ in 2020, and $32.83 \pm 1.64$ µg/m³ in 2021. The highest average monthly $SO_2$ concentration was in the year 2017 ($189.00 \pm 9.45$ µg/m³), whereas the lowest was in the year 2018 and 2019 ($0.00 \pm 0.00$ µg/m³). In 2017, the highest average monthly concentration was detected in December ($189.00 \pm 9.45$ µg/m³), whereas the lowest average monthly concentration was detected in July ($59.00 \pm 2.95$ µg/m³). In 2018, the highest average monthly concentration was detected in February and March ($62.00 \pm 3.10$ µg/m³), while the lowest average monthly concentration was detected in July, August, and September ($0.00 + 0.00$ µg/m³). Average monthly $SO_2$ concentrations at location B were $22.08 \pm 1.10$ µg/m³ in 2017, $17.08 \pm 0.85$ µg/m³ in 2018, $13.92 \pm 0.69$ µg/m³ in 2019, $7.16 \pm 0.36$ µg/m³ in 2020, and $15.67 \pm 0.78$ µg/m³ in 2021. Average monthly $NO_2$ concentration at location B was the highest in the year 2017 ($40.00 \pm 2.00$ µg/m³), whereas the lowest was detected in the year 2020 ($0.00 \pm 0.00$ µg/m³). In 2017, the highest average monthly concentration was detected in January ($40.00 \pm 2.00$ µg/m³), whereas the lowest average monthly concentration in that year was detected in June and July ($13.00 \pm 0.65$ µg/m³). In 2020, the highest average monthly concentration of $22.00 \pm 1.10$ µg/m³ was detected in January, whereas during other months, the average monthly concentration varied from $0.00 \pm 0.00$ µg/m³ to $17.00 \pm 0.85$ µg/m³. Average monthly $NO_2$ concentrations at location D were $7.00 \pm 0.35$ µg/m³ in 2017, $21.00 \pm 1.05$ µg/m³ in 2018, $13.42 \pm 0.67$ µg/m³ in 2019, $15.08 \pm 0.75$ µg/m³ in 2020, and $13.58 \pm 0.68$ µg/m³ in 2021. The highest average monthly $NO_2$ concentration was in 2017 ($30.00 \pm 1.50$ µg/m³), whereas the lowest was in the year 2019 ($5.00 \pm 0.25$ µg/m³). In 2017, the highest average monthly concentration was detected in May ($30.00 \pm 1.50$ µg/m³), the lowest average monthly concentration was detected in September that year ($14.00 \pm 0.70$ µg/m³). In 2019, the highest average monthly concentration was detected in January and February ($16.00 \pm 0.80$ µg/m³), and the lowest average monthly concentration was detected in May ($5.00 \pm 0.25$ µg/m³). Average monthly $NO_2$ concentrations at location E were $6.58 \pm 0.32$ µg/m³ in 2017, $6.67 \pm 0.33$ µg/m³ in 2018, $4.83 \pm 0.24$ µg/m³ in 2019, $5.17 \pm 0.26$ µg/m³ in 2020, and $5.58 \pm 0.28$ µg/m³ in 2021. The highest average monthly $NO_2$ concentration was in the year 2017 ($14.00 \pm 0.70$ µg/m³), whereas the lowest was in the year 2019 and 2021 ($2.00 \pm 0.10$). Again, the highest average monthly concentrations of $NO_2$ were detected in January, December, and February, whereas the lowest average monthly concentrations were detected during summer months.

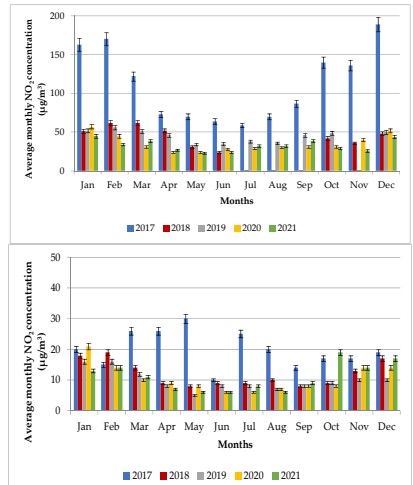
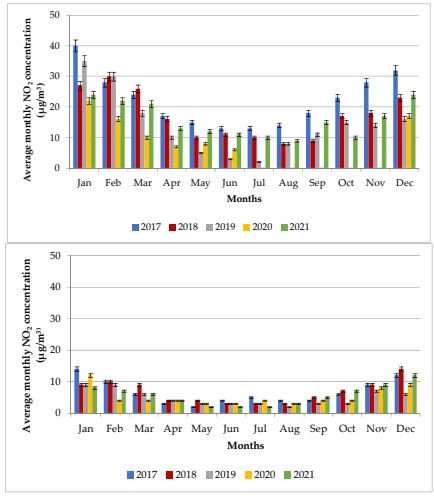

**Figure 4.** Average monthly $NO_2$ concentrations at locations A, B, D, and E between 2017 and 2021 (from left to right, from top to bottom).

Figure 5 shows average annual $NO_2$ concentrations at studied locations between 2017 and 2021. Average annual $NO_2$ concentrations were the highest at location A (50.00 ± 2.50 µg/m³) in 2017, while the lowest were detected at location E (5.00 ± 0.25 µg/m³) in 2019, 2020, and 2021.

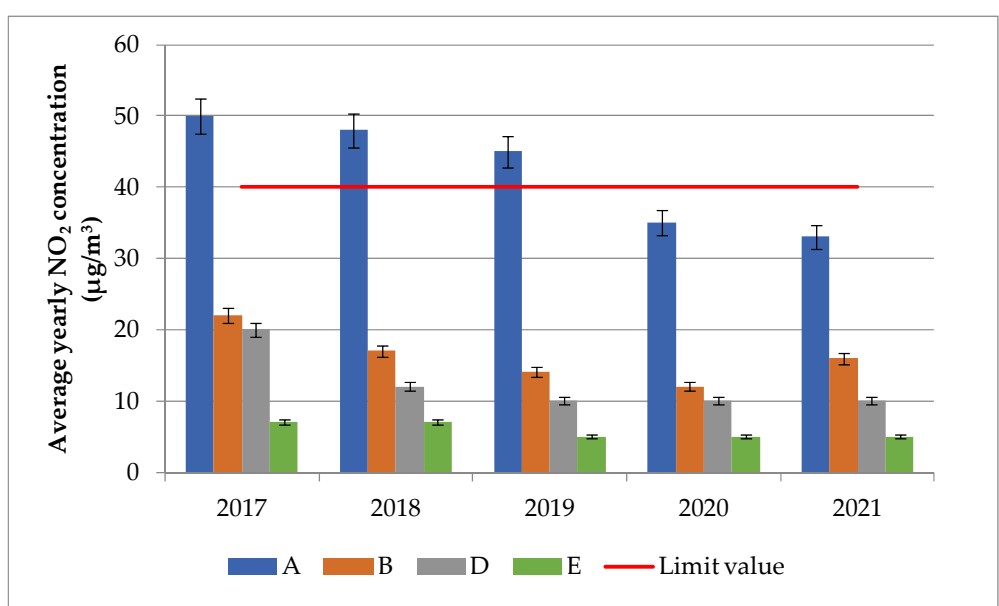

**Figure 5.** Average yearly $NO_2$ concentrations at locations A, B, D, and E between 2017 and 2021.

In addition, Figures 6 and 7 present the average monthly and annual $NO_X$ concentrations at chosen locations during studied years, and Table 2 shows average, minimal, and maximal values with standard deviation. The plots for daily average concentrations of $NO_X$ at all locations during the study period are in the SM (Figure S3). Average, minimum, and maximum values with standard deviations for all studied years for each pollutant are additionally included (SM, Table S3).

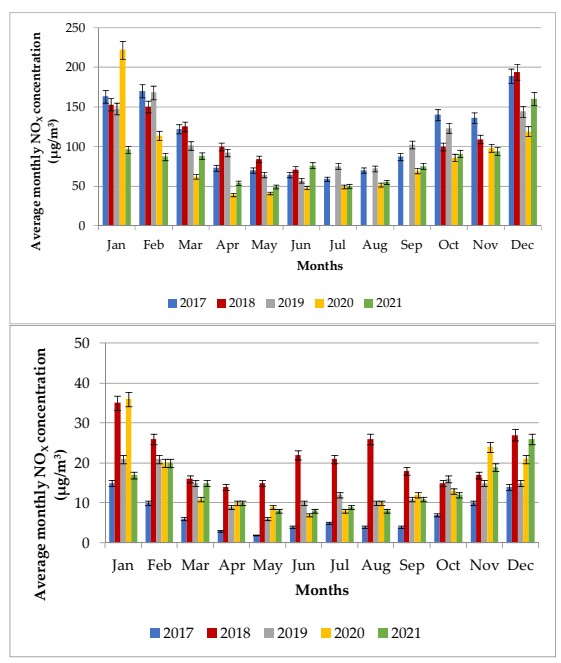
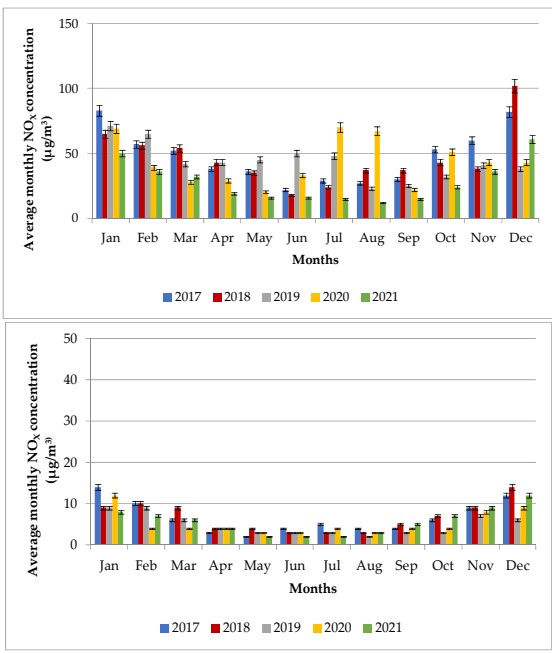

**Figure 6.** Average monthly $NO_X$ concentrations at locations A, B, D, and E between 2017 and 2021 (from left to right, from top to bottom).

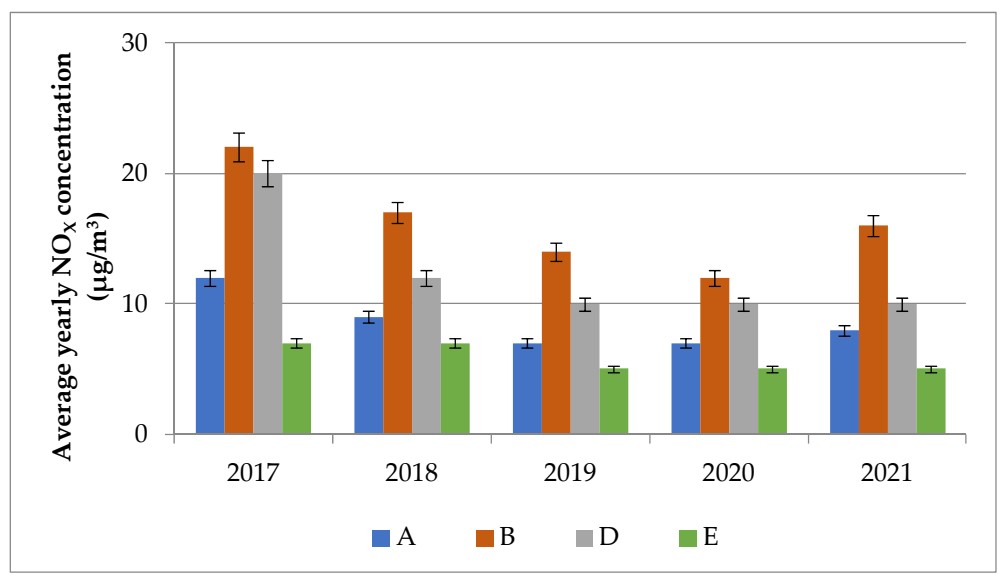

**Figure 7.** Average annual $NO_X$ concentrations at locations A, B, D, and E between 2017 and 2021.

**Table 2.** Average, minimum, and maximum annual $NO_X$ ($\mu g/m^3$) values at studied locations.

|  |  | A | B | D | E |
|---|---|---|---|---|---|
| 2017 | average | 111.92 ± 5.97 | 47.42 ± 2.37 | 7.00 ± 0.35 | 6.16 ± 0.31 |
|  | min. | 59.00 ± 2.95 | 22.00 ± 1.10 | 2.00 ± 0.10 | 2.00 ± 0.10 |
|  | max. | 159.00 ± 9.45 | 83.00 ± 4.15 | 15.00 ± 0.75 | 15.00 ± 0.75 |
| 2018 | average | 90.50 ± 4.53 | 46.00 ± 2.30 | 21.00 ± 1.05 | 7.58 ± 0.38 |
|  | min. | 0.00 ± 0.00 | 18.00 ± 0.90 | 14.00 ± 0.70 | 4.00 ± 0.20 |
|  | max. | 194.00 ± 9.70 | 102.00 ± 5.10 | 35.00 ± 1.75 | 15.00 ± 0.75 |
| 2019 | average | 90.50 ± 4.53 | 43.58 ± 2.18 | 13.42 ± 0.67 | 5.58 ± 0.28 |
|  | min. | 0.00 ± 0.00 | 23.00 ± 1.15 | 6.00 ± 0.30 | 3.00 ± 0.15 |
|  | max. | 168.00 ± 8.40 | 71.00 ± 3.55 | 21.00 ± 1.05 | 10.00 ± 0.50 |
| 2020 | average | 83.30 ± 4.17 | 42.83 ± 2.14 | 15.08 ± 0.75 | 6.08 ± 0.30 |
|  | min. | 39.00 ± 1.95 | 20.00 ± 1.00 | 7.00 ± 0.35 | 2.00 ± 0.10 |
|  | max. | 222.00 ± 11.10 | 70.00 ± 3.50 | 36.00 ± 1.80 | 15.00 ± 0.75 |
| 2021 | average | 81.30 ± 4.01 | 27.67 ± 1.36 | 13.58 ± 0.70 | 6.16 ± 0.31 |
|  | min. | 49.00 ± 2.45 | 12.00 ± 0.60 | 8.00 ± 0.40 | 2.00 ± 0.10 |
|  | max. | 160.00 ± 8.00 | 61.00 ± 3.05 | 26.00 ± 1.30 | 14.00 ± 0.70 |

$NO_2$ and $NO_X$ concentration levels have a characteristic daily and annual course [17]. The highest $NO_2/NO_X$ concentration levels were detected in 2017 at all locations, whereas the lowest were detected in the year 2020 and 2021. It is believed that changes in the average annual levels of $NO_2/NO_X$ through the years may be a consequence of meteorological conditions. At all measuring points, the lowest levels are measured in the summer months when the weather conditions for the dilution of emissions are more favorable. During this period, emissions of nitrogen oxides are also lower due to reduced traffic (permits, holidays, and an increased use of bicycles). Levels of nitrogen oxides are highest in winter when the atmosphere is the least stable and the worst ventilated, and discharges are slightly higher than in the summer [17]. Furthermore, the concentration levels in 2020 and 2021 have also been affected by the COVID-19 pandemic and measures set by the Slovenian government. In the RS, two official lockdowns occurred (12 March–31 May 2020 and 18

October–31 December 2020). During the lockdowns, the Slovenian government imposed strict measures and control policies to preserve the health and safety for Slovenian citizens, such as closing public institutions, working and schooling from home, or cancelling all public events. Furthermore, public transportation and travel in and out of the country were shut down; therefore, there was significantly less traffic (road, railway, or even aviation) [14] leading to less $NO_2/NO_X$ emissions.

### 3.3. $PM_{10}$ and $PM_{2.5}$ Concentrations between 2017 and 2021

Elevated levels of $PM_{10}$ and $PM_{2.5}$ particles typically occur in the RS in the wintertime when weak winds in the ground layers occur (so-called temperature reversal), such as in the case of anticyclone conditions. In these layers, there is a weak vertical mixing of the air, which causes the pollutants to stay close to the ground for a long time. At the same time, small fireplaces are the most active during the wintertime, with the largest contribution to $PM_{10}$ and $PM_{2.5}$ particulate emissions. Furthermore, transport is a major source of pollution with the smallest particles, especially in areas with high traffic density.

The limit value for the daily maximum average of $PM_{10}$ in the RS is 50 $\mu g/m^3$ and must not be exceeded by more than 35 times per year. The same value is proposed by the WHO [7]. The average annual limit value is set to 40 $\mu g/m^3$ in the RS, whereas the WHO proposed this value to be 20 $\mu g/m^3$ [7,17]. The limit value for the daily maximum average of $PM_{2.5}$ in the RS is not determined, whereas the WHO proposed this value to be 25 $\mu g/m^3$. The average annual limit value in the RS is 20 $\mu g/m^3$, which the WHO proposed instead to be 10 $\mu g/m^3$.

The $PM_{10}$ parameter was studied during the period from 2017 to 2021 at four locations (A, B, D), and the data availability was more than 85% at all locations. Concentration levels of $PM_{10}$ were not monitored at location E. The plots for daily average concentrations of $PM_{10}$ at all locations during the study period are in the SM (Figure S3). Average, minimum, and maximum values with standard deviations for all studied years for each pollutant are additionally added (SM, Table S3). From the daily average data, it can be observed that the $PM_{10}$ concentrations were the lowest at location C in the year 2019 ($14.95 \pm 0.75$ $\mu g/m^3$), while the highest were at location A in the year 2017 ($35.48 \pm 1.77$ $\mu g/m^3$). Location A is located very closely to a road with significant daily traffic, which coincides with the obtained average value.

Figure 8 shows average monthly $PM_{10}$ concentrations at studied locations throughout the years. Average monthly $PM_{10}$ concentrations at location A were $32.66 \pm 1.63$ $\mu g/m^3$ in 2017, $35.33 \pm 1.706$ $\mu g/m^3$ in 2018, $32.58 \pm 1.63$ $\mu g/m^3$ in 2019, $30.16 \pm 1.50$ $\mu g/m^3$ in 2020, and $29.75 \pm 1.49$ $\mu g/m^3$ in 2021. Average monthly concentrations were nearly the same throughout the studied years (< 10%). The highest average monthly $PM_{10}$ concentration was detected in January 2017 ($64.00 \pm 3.20$ $\mu g/m^3$), whereas the lowest average monthly $PM_{10}$ concentration was detected in May and August 2021 ($17.00 \pm 0.85$ $\mu g/m^3$). Average monthly $PM_{10}$ concentrations at location B were $25.08 \pm 1.25$ $\mu g/m^3$ in 2017, $28.83 \pm 1.44$ $\mu g/m^3$ in 2018, $23.08 \pm 1.15$ $\mu g/m^3$ in 2019, $21.33 \pm 1.06$ $\mu g/m^3$ in 2020, and $19.42 \pm 0.97$ $\mu g/m^3$ in 2021. The average monthly $PM_{10}$ concentration at location B was the highest in January 2017 ($67.00 \pm 3.35$ $\mu g/m^3$), whereas the lowest was detected in September 2017 ($8.00 \pm 0.40$ $\mu g/m^3$). Due to an incorrect operation in the measuring system, the measurements were not established for December 2021. Automatic continuous measurements of $PM_{10}$ at location C started in March 2018. Average monthly $PM_{10}$ concentrations at this location were therefore $17.33 \pm 0.86$ $\mu g/m^3$ in 2018, $17.08 \pm 0.85$ $\mu g/m^3$ in 2019, $15.5 \pm 0.78$ $\mu g/m^3$ in 2020, and $20.75 \pm 1.04$ $\mu g/m^3$ in 2021. Average monthly $PM_{10}$ concentration at location B was the highest in January and September 2021 ($34.00 \pm 1.70$ $\mu g/m^3$), whereas the lowest was detected in March, April, and May 2020 ($4.00 \pm 0.20$ $\mu g/m^3$). Average monthly $PM_{10}$ concentrations at location D were $19.42 \pm 0.97$ $\mu g/m^3$ in 2017, $21.50 \pm 1.08$ $\mu g/m^3$ in 2018, $18.75 \pm 0.94$ $\mu g/m^3$ in 2019, $17.83 \pm 0.89$ $\mu g/m^3$ in 2020, and $16.25 \pm 0.81$ $\mu g/m^3$ in 2021. Average monthly concentrations were nearly the same through the studied years. The highest average monthly

$PM_{10}$ concentration was detected in January 2017 ($42.00 \pm 2.10$ µg/m³), whereas the lowest average monthly $PM_{10}$ concentration was detected in May 2021 ($6.00 \pm 0.30$ µg/m³).

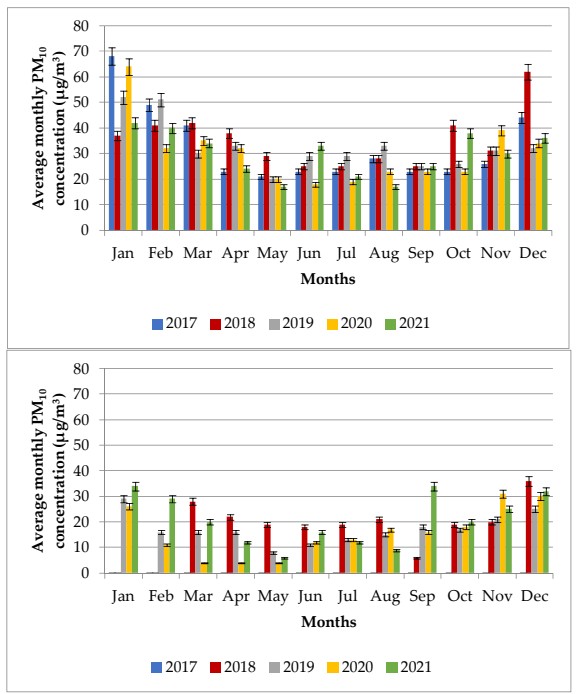
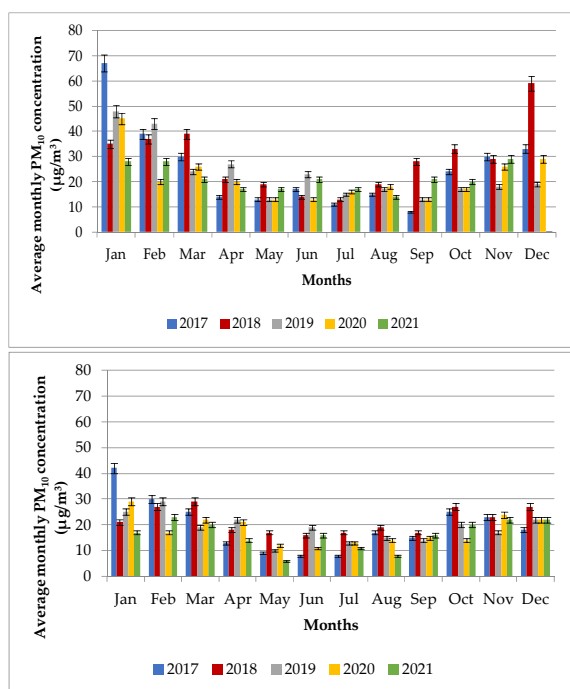

**Figure 8.** Average monthly $PM_{10}$ concentrations at locations A, B, C, and D between 2017 and 2021 (from left to right, from top to bottom).

Figure 9 additionally shows average annual $PM_{10}$ concentrations at studied locations between 2017 and 2021. Average annual $PM_{10}$ concentrations were the highest at location A ($50.00 \pm 2.50$ µg/m³) in 2017, while the lowest were detected at location E ($5.00 \pm 0.25$ µg/m³) in 2019, 2020, and 2021.

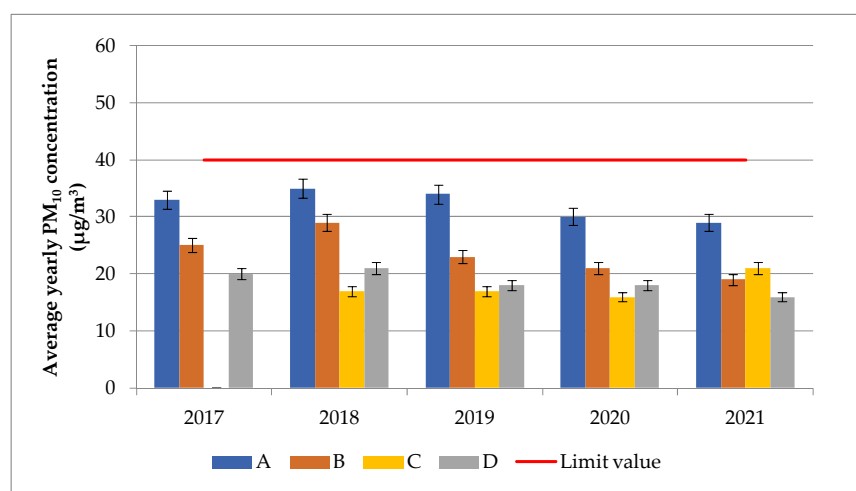

**Figure 9.** Average annual $PM_{10}$ concentrations at locations A, B, D, and E between 2017 and 2021.

The $PM_{2.5}$ parameter was studied during the period from 2017 to 2021 at only two locations (A and D). The data availability was more than 70% at both locations. At other monitoring stations, the measurements of $PM_{2.5}$ were not established. The plots for daily average concentrations of $PM_{2.5}$ at these two locations during the study period are in the SM (Figure S4). Average, minimum, and maximum values with standard deviations are additionally included (SM, Table S4).

At location A, the measurements were established in April 2018. Average monthly $PM_{2.5}$ concentrations at this location were $13.58 \pm 0.80$ µg/m$^3$ in 2018, $21.50 \pm 1.08$ µg/m$^3$ in 2019, $19.00 \pm 0.95$ µg/m$^3$ in 2020, and $14.16 \pm 0.71$ µg/m$^3$ in 2021. The highest average monthly $PM_{2.5}$ concentration was detected in January 2020 ($57.00 \pm 2.85$ µg/m$^3$), whereas the lowest average monthly $PM_{2.5}$ concentration was detected in May 2021 ($6.00 \pm 0.30$ µg/m$^3$). Due to an incorrect operation in the measuring system, the measurements were not established for December 2021. At location D, the measurements of $PM_{2.5}$ were established until 2021. Therefore, the average monthly $PM_{2.5}$ concentration in 2021 was $12.00 \pm 0.60$ µg/m$^3$. The highest average monthly concentration was detected in November and December 2021 ($20.00 \pm 1.00$ µg/m$^3$), whereas the lowest was detected in May 2021 ($2.00 \pm 0.10$ µg/m$^3$) (Figure 10). Figure 11 additionally shows average annual $PM_{2.5}$ concentrations at locations A and D between 2017 and 2021.

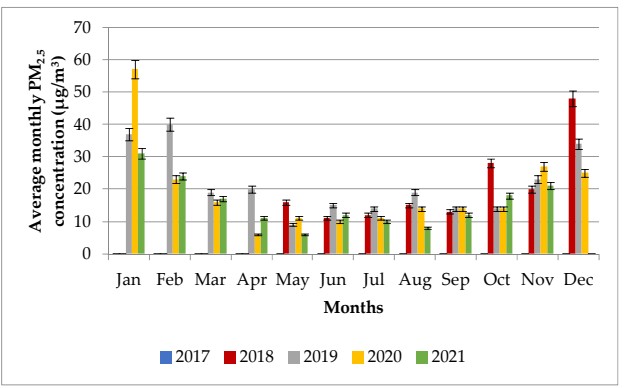
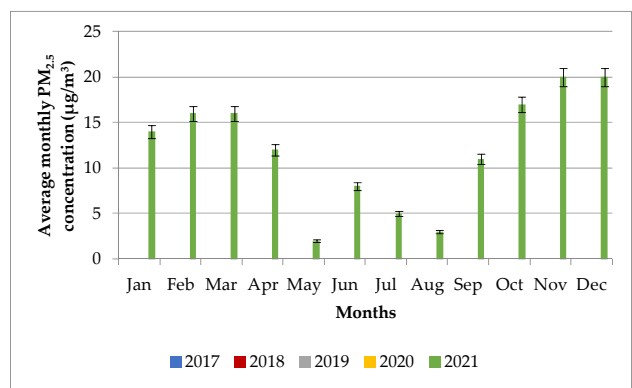

**Figure 10.** Average monthly $PM_{2.5}$ concentrations at locations A and D between 2017 and 2021 (from left to right).

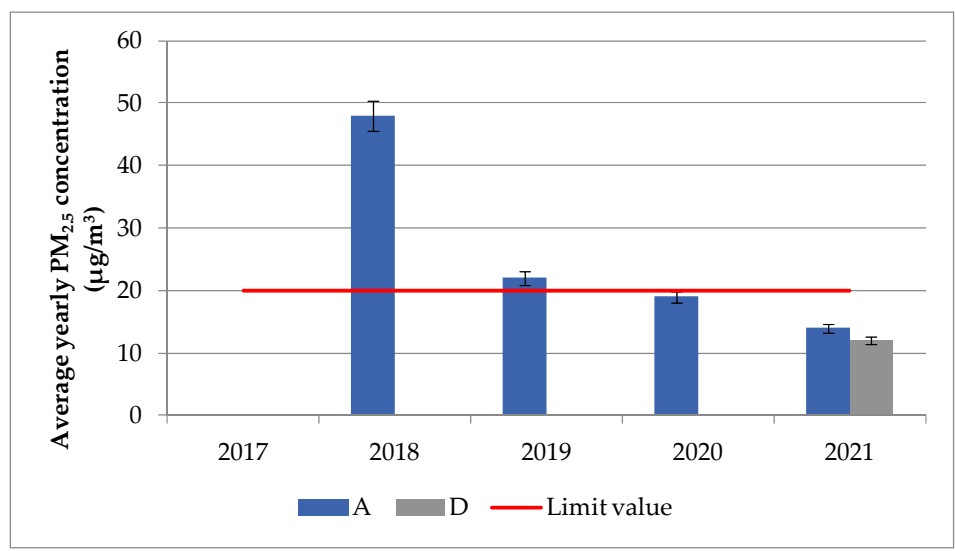

**Figure 11.** Average annual $PM_{2.5}$ concentrations at locations A and D between 2017 and 2021.

To sum up, concentration levels of $PM_{10}$ and $PM_{2.5}$ were higher during the cold period of the year compared to spring and summer seasons. The difference between autumn/winter and spring/summer concentration levels has been calculated to be around 15%. The concentration levels of $PM_{10}$ and $PM_{2.5}$ between 2018 and 2020 were already studied by the same authors in our previous work [14]. Before 2018 and 2021, the concentrations values were negligible. The year 2021 was still marked with the COVID-19 pandemic, and thus obtained values may be affected by it. It was similarly found in other works [23,24].

### 3.4. Correlation between $NO_X$ and $NO_2$ during 2019 and 2020

Table 3 shows the correlation coefficients between daily $NO_X$ and $NO_2$ parameters between the years 2019 and 2020, as these two years represent a mark in analyzing air quality in the RS due to the COVID-19 pandemic. In addition, the year 2021 was added to obtain representative results. Pearson's correlation coefficient was studied according to the following works [25,26] and results are presented based on the location and studied year. In addition, during these two years, the concentration levels of nitrogen oxides were studied between weekdays and weekends and the main findings are presented.

**Table 3.** Correlation between nitrogen oxides.

| | | 2019 | | 2020 | | 2021 | |
|---|---|---|---|---|---|---|---|
| | | $NO_2$ | $NO_X$ | $NO_2$ | $NO_X$ | $NO_2$ | $NO_X$ |
| **A** | $NO_2$ | 1 | | 1 | | 1 | |
| | $NO_X$ | +0.84 * | 1 | +0.93 * | 1 | +0.94 * | 1 |
| **B** | $NO_2$ | 1 | | 1 | | 1 | |
| | $NO_X$ | +0.73 * | 1 | +0.72 * | 1 | +0.81 * | 1 |
| **D** | $NO_2$ | 1 | | 1 | | 1 | |
| | $NO_X$ | +0.82 * | 1 | +0.92 * | 1 | +0.89 * | 1 |
| **E** | $NO_2$ | 1 | | 1 | | 1 | |
| | $NO_X$ | +0.80 * | 1 | +0.70 * | 1 | +0.72 | 1 |

* statistically significant at level <0.05.

From the obtained results, it can be observed that $NO_2$ and $NO_X$ have a positive correlation coefficient between each other. At location A in 2019, 2020, and 2021, the correlation coefficients were found to be 0.84, 0.93, and 0.94. Similarly, at location B, those values were found to be 0.73 in 2019, 0.72 in 2020, and 0.81 in 2021. At location D in 2019, 2020, and 2021, the correlation coefficients were found to be 0.82, 0.93, and 0.89, and finally, at location E, they were 0.80 in 2019, 0.69 in 2020, and 0.72 in 2021. The results are similar to the ones in the literature [27,28] and previously mentioned works, and they imply a significant correlation between the studied parameters.

Furthermore, in SM, Figure S6 shows the daily course of the average hourly $NO_2$ concentration levels at measuring points during 2019 and 2020 to determine the difference in concentration levels between weekdays and weekends. The daily trend of $NO_2$ shows that the first period of higher concentration levels occur at all measuring points during the morning traffic rush due to increased emissions from traffic. In the afternoon or in the evening, another increase occurs as a result of the atmosphere calming down in the layer of air near the ground, which is more or less pronounced at different measuring points. The highest concentration levels were detected at measuring location A, which is classified as traffic location. The lowest concentrations were detected at measuring location E. On weekdays, the levels are higher for most of the day due to more intense traffic, while on weekends, the highest values were recorded in the evening hours. Other works confirmed similar conclusions [29,30].

### 3.5. Correlation between $PM_{10}$ and $PM_{2.5}$

The term particle matter is a general term that means suspended particles (liquid solids) in gas. With the term $PM_{2.5}$, fine particles that have an aerodynamic diameter of less than 2.5 μm are described, and with the term $PM_{10}$, particles with an aerodynamic diameter of less than 10 μm are described. Regarding the origin, the particles can be primary and secondary, natural or of anthropogenic origin. Natural sources are primarily a consequence of the intake of sea salt, natural re-suspension of oils, desert dust, and flower dust, whereas

anthropogenic sources include emissions not related to fuel combustion in thermal energy facilities and industry, with heating of apartments and other buildings, and traffic [5].

The correlation coefficient between $PM_{10}$ and $PM_{2.5}$ was studied based on the work of Talbi et al. [31], Usman et al. [32], and Truong et al. [33], again between the years 2019, 2020, and 2021. Obtained results are presented based on the location and studied year. In addition, during these two years, the concentration levels of PM were studied between heating seasons, and the main findings are presented.

The correlation coefficients during the study period are shown in Table 4. At location A in 2019, the correlation coefficient was 0.95, while at location B was a little lower, at 0.92. In 2020, the correlation coefficient at location A was 0.94, the same was found in the year 2021. At location B, the correlation coefficient was 0.92 in 2020 and 0.77 in 2021.

**Table 4.** Correlation coefficient between the PM fractions.

| | | 2019 | | 2020 | | 2021 | |
|---|---|---|---|---|---|---|---|
| | | $PM_{10}$ | $PM_{2.5}$ | $PM_{10}$ | $PM_{2.5}$ | $PM_{10}$ | $PM_{2.5}$ |
| A | $PM_{10}$ | 1 | | 1 | | 1 | |
| | $PM_{2.5}$ | +0.957 * | 1 | +0.94 * | 1 | +0.94 * | 1 |
| D | $PM_{10}$ | 1 | | 1 | | 1 | |
| | $PM_{2.5}$ | +0.92 * | 1 | +0.92 * | 1 | +0.77 * | 1 |

* statistically signifant at level <0.05.

From the obtained results, it can be concluded that particulate matter have a positive and significant correlation coefficient between each other.

In SM, Figure S7 shows the daily course of the average hourly $PM_{10}$ and $PM_{2.5}$ concentration levels at locations A, C, and D during the heating season of 2019 and 2020. Heating season is between October and March the following year. The highest concentration levels were detected during morning and evening hours at all locations. The evening maximum is more pronounced, when the traffic rush is joined by discharges due to heating, but at the same time, a temperature reversal begins to form in the evening and consequently decreases the dilution of polluted air.

### 3.6. Meteorological Conditions

Jiaxin et al. [34] stated that air quality depends on pollution emissions and regional meteorological conditions. Changes in meteorological conditions can affect the physical processes and chemical reactions of pollutants [35,36]. Wind speed and direction can affect the diffusion and transmission of pollutants, while temperature and humidity can affect pollutant reaction processes [14].

Meteorological data were studied in details between the years 2019 and 2020 to better understand the impact of metrology on the concentration levels of studied air pollutants. According to the meteorological data, the mean annual average temperature in 2019 on location A was 12.66 ± 0.63 °C, on location B 12.61 ± 0.63 °C, on location C 12.14 ± 0.61 °C, on location D 11.62 ± 0.58 °C, and on location E 22.33 ± 1.12 °C. The average annual relative humidity (RH) was 65.94 ± 3.29%, 78.46 ± 3.92%, 80.91 ± 4.05%, respectively. In the year 2020, the average annual temperature was 12.21 ± 0.61 °C, 12.04 ± 0.60 °C, 11.65 ± 0.58 °C, 10.76 ± 0.53 °C, and 23.50 ± 1.18 °C, respectively, at each location, whereas RH was 64.25 ± 3.21%, 77.08 ± 3.85%, 80.41 ± 4.02%, respectively. On location C and E, RH was not monitored during the studied years. In SM, Figure S6 shows the plots of temperature and RH for studied years at each location. Moreover, Figure 12 shows wind roses for each location and each studied year. As stated in our previous work [14], the wind conditions in the RS are influenced by its varied relief, location, and the Alps. Normally, the wind blows from a westerly direction. Compared to Western Europe, the RS is not as windy, as it is located in the lee of the Alps [3]. Average

wind speeds at location A were $1.30 \pm 0.07$ m/s in 2019 and 2020, at location B were $1.70 \pm 0.85$ m/s in 2019 and 2020, at location C were $0.90 \pm 0.05$ m/s in 2019 and $1.00 \pm 0.05$ m/s, and at locations D and E were $1.50 \pm 0.08$ m/s in 2019 and 2020. During both years, north-east winds dominated location A, east winds dominated location B, north-west winds dominated location C, and east winds dominated both locations D and E. Based on these findings, it can be concluded that changes in temperature, relative humidity, wind speed, and wind direction were negligible at all locations during both studied years (<5%).

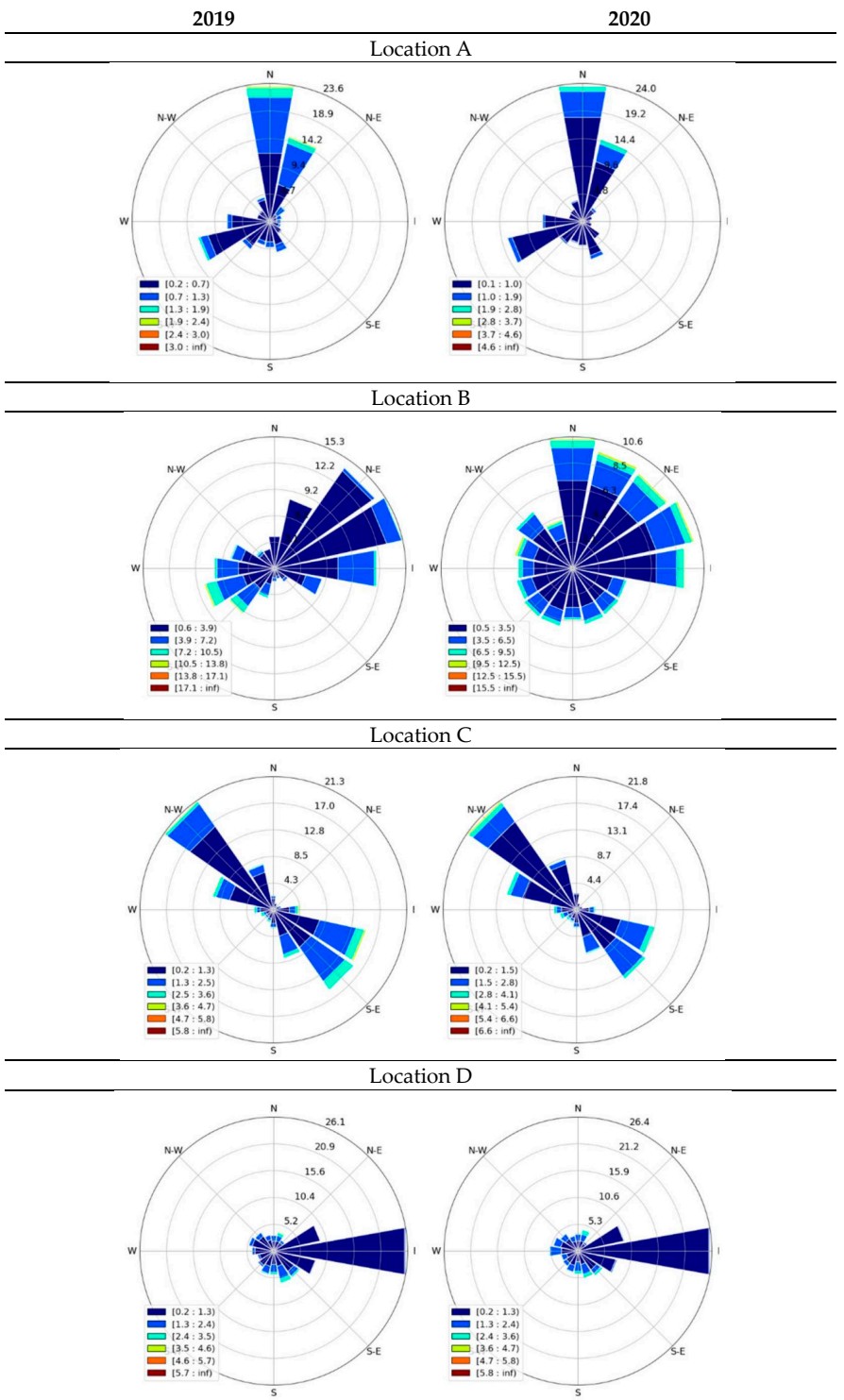

**Figure 12.** *Cont.*

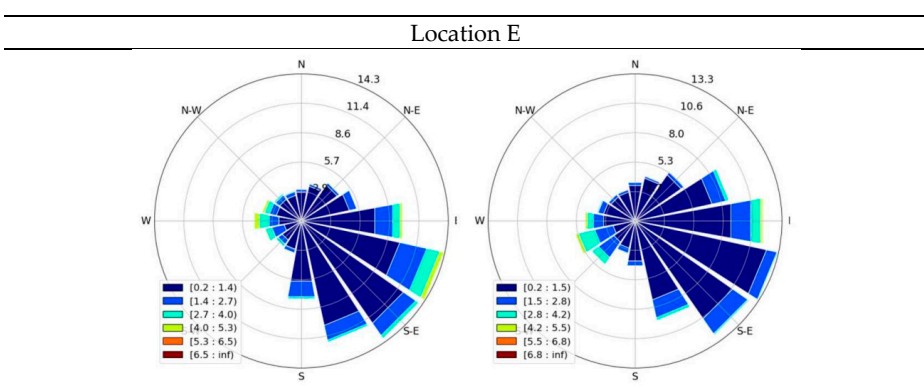

**Figure 12.** Wind roses for each studied location and year.

*3.7. Emission Sources in the Republic of Slovenia*

Today, the largest sources of $SO_X$ in the RS are still the production of electricity and thermal energy, industrial processes, and the use of fuels in industry. In the past, the largest source was the use of coal for heating households. The annual $SO_X$ emissions in the RS in 2020 were approximately 4.0 thousand tons, and, compared to the year 1980, those emissions decreased by as much as 98% [17]. The reduction of discharges is mainly due to the installation of desulfurization devices in thermal power plants, the use of coal with a lower sulfur content, and the replacement of liquid and solid fuels in industry with natural gas. The largest share of total SOx emissions in 2020 was contributed by thermal power plants and heating plants, which was 37%, followed by industrial processes with 32% [17].

Annual emissions of $NO_X$ in the RS are roughly 34 thousand tons [17]. By the year 2019, they decreased by approximately 60%. The largest source of $NO_X$ emissions represented road traffic as it contributed as much as 47% to the total national emissions. $NO_2$ concentrations have characteristic annual and daily levels. At all measuring points, the lowest levels are measured in the summer months when the weather conditions for the dilution of emissions are more favorable. During this period, emissions of nitrogen oxides are also lower due to reduced traffic. Levels of $NO_2$ are usually the highest in winter when the atmosphere is the worst ventilated. During the morning rush hour, the levels of $NO_2$ are the highest due to increased emissions from traffic, and the same can be stated for the "after work" rush hours. Therefore, on weekdays, the values are the highest during the day due to the more intensive traffic, while during weekends, the highest values are recorded in the evening. The year 2020 was marked by the COVID-19 pandemic. Due to the movement restriction, home working and schooling were restricted, and road traffic subsequently decreased. Especially in March and April, the $NO_X$ emissions from traffic decreased throughout Europe, and, consequently, $NO_2$ levels decreased in urban and suburban environments [14,24]. The $NO_2$ concentrations in the RS were at the lowest measuring point in 2020 compared to previous years (<50 wt.%).

Furthermore, the epidemiological studies show that $PM_{10}$ and $PM_{2.5}$ have the most negative impact in terms of air pollution [11,37]. Reports from the WHO indicate that there is no limit below which no health effects are expected [20,38]. The impact on health is caused by the inhalation of particles and resulting invasion into the lungs and blood, which causes damage to the respiratory, cardiovascular, immune, and nervous systems [37]. The smaller the particles, the deeper they can penetrate into the lungs until inflammation or tissue damage occurs due to both chemical and physical interactions between the particles and the tissue. In addition to the negative impact on health, particle pollution has an impact on the climate and ecosystem [16]. Annual emissions of $PM_{10}$ particles in 2019 amounted to 13 thousand tons, and annual emissions of $PM_{2.5}$ were approximately 11 thousand tons. It has been stated by the ARSO [5] that in the period 2000–2019, emissions of $PM_{10}$ and $PM_{2.5}$ were reduced by 25%. The reduction in emissions is the result of the improvement of energy efficiency and combustion processes, the modernization of technological processes,

the replacement of solid fossil fuels with natural gas and renewable energy sources, and the introduction of stricter emission regulations standards for motor vehicles. In 2019, small fireplaces caused around 60% of emissions at the national level, $PM_{10}$ and $PM_{2.5}$ emissions from road traffic represented 8% and 7% of total national emissions, respectively, and $PM_{2.5}$ emissions were 7%. In March 2020, those values were generally lower. In 2020, a high amount of Sahara dust came across the country, causing the concentrations of $PM_{10}$ and $PM_{2.5}$ to jump higher than 100 $\mu g/m^3$.

## 4. Conclusions

From the daily average data, it can be observed that $SO_2$ concentrations were the lowest in the year 2021 at location B, which is classified as background location, while the highest were detected in the year 2018 at location E, which is also classified as background location. The average daily concentrations of $NO_2$ and $NO_X$ daily average were the highest at location A in the year 2017, whereas the lowest were detected in the years 2010 and 2021. It is believed that those results are a consequence of measures set by the Slovenian government during the COVID-19 epidemic. The $PM_{10}$ and $PM_{2.5}$ daily average concentrations were the highest at location A in 2017, and the lowest were observed in the year 2019 at location C. In total, the daily, monthly, and annual concentrations for all the studied parameters decreased from 2017 to 2021 at all studied locations, even though some anomalies occurred. There are several reasons for such results: updated laws and standards in the EU and the RS, meteorological conditions, the COVID-19 pandemic and its set measures, natural sources such as Saharan dust, etc. Furthermore, the results from meteorological data showed a connection to the concentrations obtained from all the air pollutants. The average annual temperatures were around 12 °C at all locations during all studied years, average wind speeds were around 1.30 m/s at all locations during all studied years, while the average relative humidity was approximately 75%. The correlation coefficients showed a link between the $NO_2$ and $NO_X$ parameters and between the $PM_{10}$ and $PM_{2.5}$ parameters, respectively.

The obtained data showed a decrease in average concentrations for all the studied parameters, confirming the positivity of the measures set by the Slovenian government. The potential sources of pollution are still traffic (mostly road and railway), agricultural residue burning, coal, biomass, and industry; therefore, the obtained data could be used in the future when national strategies regarding air quality are updated and improved. In the future, the work will focus on analyzing the quartile and seasonal changes in concentration levels of studied parameters, as well meteorological parameters, which will include levels of rain fall during the study period. The correlation between meteorological conditions and particulate matter will be studied in detail in addition to ozone and VOCs concentration levels.

**Supplementary Materials:** The following supporting information can be downloaded at: https://www.mdpi.com/article/10.3390/atmos14030578/s1, Figure S1: Average daily $SO_2$ concentrations at locations (A–E) between 2017 and 2021; Figure S2: Average daily $NO_2$ concentrations at locations (A–E) between 2017 and 2021; Figure S3: Average daily $NO_X$ concentrations at locations (A–E) between 2017 and 2021; Figure S4: Average daily $PM_{10}$ concentrations at locations (A–E) between 2017 and 2021; Figure S5: Average daily $PM_{2.5}$ concentrations at locations (A–E) between 2017 and 2021; Figure S6: Daily course of the average hourly $NO_2$ concentration levels at measuring points between 2019 and 2020; Figure S7: Daily course of the average hourly $PM_{10}$ concentration levels during the heating season 2019 and 2020; Figure S8: Plots of temperature and RH at locations (A–E) between 2019 and 2020; Table S1: Average, minimum, and maximum daily $SO_2$ concentrations ($\mu g/m^3$) at all locations between 2017 and 2021; Table S2: Average, minimum, and maximum daily $NO_2$ concentrations ($\mu g/m^3$) at all locations between 2017 and 2021; Table S3: Average, minimum, and maximum daily $NO_X$ concentrations ($\mu g/m^3$) at all locations between 2017 and 2021; Table S4: Average, minimum, and maximum daily $PM_{10}$ concentrations ($\mu g/m^3$) at all locations between 2017 and 2021; Table S5: Average, minimum, and maximum daily $PM_{2.5}$ concentrations ($\mu g/m^3$) at all locations between 2017 and 2021.

**Author Contributions:** Conceptualization, methodology, software, validation, investigation, formal analysis, writing—original draft: M.I.; software, validation, investigation, writing—original draft K.A.; methodology, validation, formal analysis, writing—review and editing, supervision: D.U.; supervision: M.S., supervision: D.G.; supervision: R.V. All authors have read and agreed to the published version of the manuscript.

**Funding:** This research received no external funding.

**Institutional Review Board Statement:** Not applicable.

**Informed Consent Statement:** Not applicable.

**Data Availability Statement:** Not applicable.

**Conflicts of Interest:** The authors declare no conflict of interest.

**Abbreviations**

| | |
|---|---|
| ARSO | Slovenian Environment Agency |
| EEA | European Environment Agency |
| EU | European Union |
| EIMV | Elektroinštitut Milan Vidmar |
| NIJZ | National Institute for Public Health |
| SM | Supplementary Materials |
| RS | Republic of Slovenia |
| WHO | World Health Organization |

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
