# Peer review of "Assessment of Air Pollution in Different Areas (Urban, Suburban, and Rural) in Slovenia from 2017 to 2021"

_atmosphere, doi:10.3390/atmos14030578_

Round 1

Reviewer 1 Report (Previous Reviewer 1)

An analysis of air pollutants (SO2, NO2, NOX, PM10, and PM2.5) in the Republic of Slovenia between the years 2017 and 2021

I have read this manuscript for the 3rd time. It has been revised better than before. The manuscript added the results of years 2017, 2018 and 2021. On the other hand, the manuscript also added the results of 3 other measurement locations representing urban areas (A and B location), suburban areas (C and D location) and rural areas (E location), instead of just 2 areas as before. The results are better analyzed and more detail presented. However, I still have some comments to contribute for a better manuscript.

1.    Title of the manuscript, which looks common, should be changed. The suggestion is “Assessment of air pollution in different areas (urban, suburban and rural) in Slovenia from 2017 to 2021”.

2.    C location missed a lot of data, so it is recommended to remove C location in the result and still have enough 3 types which represent for urban, suburban and rural and also represent for traffic, background and industry.

3.    Why did only correlations of years 2019 and 2020 evaluate? It should evaluate for all years from 2017-2021. The correlation between NOx and NO2 is low. The R2 values in figure 13 and 15 are only 0.0419 and 0.2254 (in 2019); 0.2084 and 0.4704 (in 2020). I still have a question: How to calculate the correlation? I could not find the correlation value which has been cited in the [40] and [41] references. I only found NO2/NOx = 0.779 [1] and 0.889 [2]. The correlation values change following the time and depend on each location. Therefore, they are not so meaningful.

Does p < 0.05 in table 4 correspond to precision up to 99.5%?

4.    Similar above, please self-check P.M correlation. Thermal power plants use charcoal which is a source of PM 2.5 pollution, please read and cite the reference [3]. How about p in this case of correlation?

5.    It needs the small discussion of different measurement zone types: traffic, industrial and background and different area types: urban, suburban and rural.

6.    It is necessary to carefully check the citation of references. For example, the sentence in lines 33-36: “According to the European Environmental Agency (EEA) and the World Health Organization (WHO), in the last decade air pollution became the second biggest environmental issue, just right after climate change [2]” cited [2] {European Commission Roadmap 2050. Policy 2012, 1–9, doi:10.2833/10759}, which is not suitable. Please use the original articles.

Another example, sentence in lines 32-33: “It represents a major global threat that is practically impossible to avoid, especially in developing countries [1]” can be cited from many references, because this is general.

Minor comments:

-        Figure 1 has 2 pictures, but no legend. The land boundary of Slovenia goes out to the sea. Please check and use the Mapinfo software to remove other layers.

-        Figure 3 should normalize at 30 ug/m3.

-        Color of the chart should be changed because the colors look similar between A and E location and between B and D location.

-        Figure 7 should be normalized.

-        Please check the format and information of all references. This manuscript cited many national articles such as: [7], [21-26], [34] which could not be found on the internet. Therefore, I could not check. Please cite the original documents instead of web links, documents from 9-14.

Conclusion:

Overall, the manuscript also added results for more data and more discussion. However, the revision is still incomplete. The results are more complementary but lacking in argument. More careful revising is required before it can be published in the journal Atmosphere.

References:

1.    Lijun Wang, Ju Wang, Xiaodong Tan, Chunsheng Fang, 2020, Analysis of NOx Pollution Characteristics in the Atmospheric Environment in Changchun City, Atmosphere11(1), 30

2.    Khaled Gasmi, Abdulaziz Aljalal, Watheq Al-Basheer, Mumin Abdulahi, 2017, Analysis of NOx, NO and NO2 ambient levels as a function of meteorological parameters in Dhahran, Saudi Arabia, Air Pollution XXV, WIT Transactions on Ecology and The Environment, Vol 211, 77-86.

3.    An Ha Truong, Minh Thuy Kim, Thi Thu Nguyen, Ngoc Tung Nguyen, Quang Trung Nguyen, 2018, Methane, Nitrous Oxide and Ammonia Emissions from Livestock Farming in the Red River Delta, Vietnam: An Inventory and Projection for 2000–2030, Sustainability10(10), 3826.

Author Response

Reviewer 2 Report (Previous Reviewer 2)

   1.       Graphs 2, 8, 10, 18 are too small, it is better to increase them

   2.       Why in figure 3 the axis is up to 60 if the data is not higher than 10? Can not see anything.

   3.       Why in Fig. 7 there are 5 indicators in the legend, and only in the graph?

   4.       Figure numbering not understood - Figure S1 (line 237), Figure S2 (line 325), Figure S3 (line 380, 431), Figure S4 (line 475), Figure S6 (line 536, 600), Figure S7 (line 576)

   5.       The formatting of references does not correspond to the requirements of the journal. See template

Round 2

Reviewer 1 Report (Previous Reviewer 1)

The revision is better. So accepted

Reviewer 2 Report (Previous Reviewer 2)

Accept in present form

This manuscript is a resubmission of an earlier submission. The following is a list of the peer review reports and author responses from that submission.

Round 1

Reviewer 1 Report

“An analysis of air pollutants (SO2, NO2, NOX, PM10, and PM2.5) in the Republic of Slovenia in years 2019 and 2020”

Overall comments:

Air pollution has always been an issue that can have a significant effect on human health which has been of interest to scientists around the world. This study was carried out by collecting data from two continuous, automatic air pollution monitoring stations located in Slovenia. The first station is located in the region of Notranjska, the second one is located in the valley along the Paka river. The measurement point for the first station is considered to represent the urban area with high traffic and the second measurement point is considered to be representative of the industrial area. The concentrations of five basic air pollutants SO2, NO2, NOX, PM10, and PM2.5 were collected from January 2019 to December 2020 at both monitoring stations. In addition, meteorological data on daily average temperature, relative humidity, wind direction, and wind speed during the study period were obtained from the EIMV and ARSO meteorological stations.

The results were evaluated by using the average hourly measured data to provide arguments on the following: 1/ Air pollutant concentrations in 2019 and 2020; 2/ Meteorological factors; 3/ Correlation between NOX and NO2; 4/ Correlation between PM10 and PM2.5; 5/ Correlations between air pollutants and meteorological conditions; 6/ Emission sources. The variation in concentrations of pollutants is discussed and explained in detail by different reasons presented in the article.

However, the results of this study appear to be very related to research by the same author on: Improvement of Air Quality during the COVID-19 Lockdowns in the Republic of Slovenia and its Connection with Meteorology. In particular, it is said that in 2020 there will be a reduction in air pollution due to COVID-19 Lockdowns. On the other hand, the previous study was more elaborate with data at 24 stations for PM10 and 5 stations for PM 2.5, with a longer time period in 2018-2020.

General discussions and questions:

1.     Please describe how many automatic continuous air monitoring stations are installed in Slovenia. Is it possible that there are only 2 stations, so the author can only collect the data from 2 stations? When were these monitoring stations installed? Why does the author not collect the data of the later times of 2021 and 2022? In addition to the 5 parameters presented in the study, there are other parameters such as O3; CO, CO2, VOCs, why not present it in a research manuscript or does the author want to separate it in another one?

2.     Please describe in more detail the operation of the measuring devices in the automatic continuous monitoring station. Especially how they come to instrument calibration to get accurate measurements.

3.     Describe the collection of meteorological data, from how many stations and from which stations? Are the meteorological stations at the same location as the air environment monitoring station?

4.     European countries are located very close to each other and have strong industrial production, so transboundary air pollution should be mentioned in the research? How to remove the effect of transboundary air pollution can draw conclusions about air pollutants in the Republic of Slovenia. Should refer to more documents: “H. Klein, M. Gauss, S. Tsyro, Asia. Nyíri, D. Heinesen, and H. Fagerli, Transboundary air pollution by sulfur, nitrogen, ozone and particulate matter in 2020, Data Note 2022”

Specific comments:

1.     Introduction: presented quite long with a lot of rambling content. This section should be focused on the main issues relevant to the research. For example, lines from 78 to 82, the ARSO's automatic continuous air environment monitoring system should be more clearly introduced. The presentation of some similar studies should refer to the European studies instead of Chinese ones, as the lines 96-113. The author's own research on: “M. Ivanovski, P. D. Lavrič, R. Vončina, D. Goričanec, and D. Urbancl, 2022, Improvement of Air Quality during the COVID-19 Lockdowns in the Republic of Slovenia and its Connection with Meteorology, Aerosol Air Qual. Res., 22, 9” needs to clarify the non-duplication for reference.

Tables 1 and 2 are presented but not discussed. It is necessary to shorten these two tables. Table 1 compares the WHO and RS standards, but important differences should be pointed out. Table 1 need not show limit values of O3, Benzene because these two parameters were not studied in this manuscript. Table 2 should not be presented in this manuscript because it has little to do with this manuscript.

Document No. [9] [EEA, emissions,” 2022. https://www.eea.europa.eu/ims/emissions-of-the-main-air. (accesed on November 11, 2022)] needs to be re-standardized for the origin and information. The title of the document should be cited instead of the website address and please check the year of publication. This is an important reference but rarely cited document in this study, lines from 47-50.

2.     Materials and methods: should be changed into methodology.

Although this presentation has been greatly changed from the author's own previous research, there are still many overlaps. Line 119, “2.1. study area” needs a description of the smaller area where the two monitoring stations are located, rather than a description of the RS country. A comparison of Notranjska region with 163.8 km2 and 300,000 people and site B with 3.7 km2 and 3,000 inhabitants does not reveal differences in population densities.

Why in table 3, column Area type is the same for “Urban”?

Should describe more details about the meteorological data collection. Data is collected from how many stations, which stations?

Table 4 should be deleted.

How to remove the false data? What is the data processing of this research method?

3.     Results:

The results should be presented for a general description first as the variations within: a daily average, then a week average, a month average, a season average, a quarter average, and a year average. According to the daily variation, it is possible to show the average concentrations of the hours, for example, the SO2 and NOx emissions during rush hours will be more than at night. The pollutant concentrations on weekdays will be higher than weekends. In this study, only the average results of the days of the year are presented, so it is difficult to track and not meaningful. The author could not explain why there were some points where the pollutant concentration increased dramatically.

Comparisons between 2020 and 2019 also need to be for the same time of year, rather than an average for the whole year. Tables 5 and 6 should be combined into 1 chart, similarly tables 7 and 8 should be.

The wind rose should be presented at least following the months of the year. Does the author consider the results of measuring the concentration of pollutants on a rainy day? This value can be used to calibrate the devices.

For correlation, I could not see the values of 0.84 and 0.93 in figure 4 and 5 and similar in figure 6 and 7. Should be normalized at the same value of coordinate axes. The charts need to be presented more beautifully, logical and scientific.

The correlation of NO2 and NOx; PM 2.5 and PM10 should be compared by time of day, week, month, quarter and then year.

In the study only the correlation of PM 2.5 and PM 10 and meteorological conditions was mentioned, so the word “air pollutants” in line 369 should be changed into PM fractions. Tables 11 and 12 show that as rainfall increases, the values of PM 2.5 and PM 10 increase accordingly, which is incorrect and unreasonable.

The sentence in the lines from 338 to 390: “Today, the largest sources of SOX in the RS are the production of electricity and thermal energy, industrial processes, and the use of fuels in industry, in the past the biggest source was the use of coal for heating households”, which contradicts the statement “Years ago, SO2 has been declared as the biggest environmental problem in EU, caused by the production and….”, lines 43-47.

4.     Manuscript cited a lot of Slovenian references which should be translated into English. Reference names should be cited instead of links. There are many documents in the reference section such as: 8 -14; 26-29… The author needs to edit the reference carefully.

Conclusions:

In general, this study is quite simple, just by collecting results from air monitoring stations with data sets of 2019 and 2020. The results are collected in 2 for automatic continuous monitoring stations, which are in the short time. On the other hand, 2020 is the special year of COVID-19 Lockdowns, so there are many fluctuations. The author needs to collect data from previous and subsequent years. The arguments in the paper are unconvincing.

Reviewer 2 Report

1. It is unclear why the research was done for 2019 and 2020. And how the results compare to previous years

2. In table 1 too long text in the 2nd column makes it very difficult to receive values. Can you present it in a different way?

3. Table 2 is better arranged in chronological order - in order of increasing years in column 2.

4. It is worth introducing the "Discussion" chapter separately

5. The conclusions are too trivial - it was noted that road and rail traffic pollute the air. Authors must indicate what is new, interesting and important in their research.

6. References does not meet the requirements of the journal. See the template

Round 2

Reviewer 1 Report

“An analysis of air pollutants (SO2, NO2, NOX, PM10, and PM2.5) in the Republic of Slovenia in the years 2019 and 2020”

After reading the revised manuscript and the cover letter of author response, I still find that this study is still quite simple, just by collecting results from air monitoring stations with data sets of 2019 and 2020. The results are only collected by internet from 2 for automatic continuous monitoring stations, which are in the short time. The explanation is not completely convincing. Although the article has been improved, but it still needs to be revised.

General discussions and questions:

1.     The answer about the choosing of data from these two monitoring stations is not convincing, because there are 25 stations, but only 2 stations are selected, 13 measuring stations representing the urban area are selected. 1 measuring station of 1 urban area. The data collection for 2 years 2019 and 2020 is quite small. It is too few data points to represent the research area.

2.     The author has not answered how to calibrate the measuring device. During a measurement day, the measuring device will need to calibrate automatically, so this time the data will be abnormal, possibly very high and very low. Therefore, the measurement data during the calibration period should be removed.

3.     It is necessary to describe the meteorological data also collected on the same website with air pollution monitoring station.

4.     The author has no discussion for the impact of transboundary air pollution? The question “How to remove the effect of transboundary air pollution?” still unanswered.

Specific comments:

1.     The introduction section is still presented quite long. Should be reduced to less than 2 pages. I don't understand why in the beginning the author put the "Literature review" section (line 96), the introduction is already the same as the general overview?

2.     Since the author did not conduct sampling, the item “2.2. Data sources and sampling process” (line 142) needs to delete the “sampling process”. It can be changed into data sources and collection. Meteorological data should be descriptive to show that the observed data are the same station and are collected on the same website.

The explanation only says “False data is not taken into account” but still doesn't answer the question “How to remove the false data? What is the data processing of this research method?”

3.     Many of my suggestions for presenting the results have not been properly expressed. For example: the daily average results should be presented in the figure with chart of hourly average. Please see to learn the presentation in the figure 2(b) of this research paper: https://www.researchgate.net/publication/362800909_Research_on_the_Influence_of_Weather_Patterns_on_Ozone_Concentration_A_Case_Study_in_Tianjin/figures?lo=1. Or the presentation in this website: https://www.kenelec.com.au/projects/air-quality-impact-monitoring/. Or like figure 10 of the article: “Experience from Integrated Air Quality Management in the Mexico City Metropolitan Area and Singapore, Atmosphere, 2019”. Similar presentation for weekly, monthly, and quarterly charts. On the other hand, also compare the same times in 2019 and 2020. Graphical presentation makes the results more attractive.

4.     I think the figures for 2021 and 2022 have been published. You can look it up here: https://www.arso.gov.si/zrak/kakovost%20zraka/podatki/

Additional comments:

1.     The charts in Figure 1 need to be re-presented for more beautiful and more scientific.

2.     Some results tables (eg: table 2-5) should be presented graphically. Please delete 219 before NO2 in table 4.

3.     Charts of Figures 2 and 3; Figures 4 and 5 need to be re-presented for more beautiful and more scientific. Take a look at the data in figure 2, some of them are on the vertical or horizontal axis respectively with NOx = 0 or NO2 = 0. I think these are auto-calibrated instrument values, if so, these values should be removed.

4.     2020 present a »COVID-19« year. However, the lockdown times are not for all year. Please describe what the lockdown times in Republic of Slovenia. And in manuscript please compare only the lockdown times in 2020 and normal times in 2019.

5.     There are still many references that have not been cited in the original documents instead of citing the website link as documents from 8-14. Line 598 has 2 words WHO. Some of self-citation should be careful, notably the reports in Slovenia language. The reference [32] has been not cited.

Conclusion:

This manuscript should be carefully revised. I think the author should spend more time to study the measurement data and discuss more carefully before it can be published. On the other hand, the author should not only focus on 2019 and 2020 but should also expand to other years such as 2018 and 2017 that can have good reliability and repeatability.

Reviewer 2 Report

Accept in present form
